# The resistomes of rural and urban pigs and poultry in Ghana

Emilie Egholm Bruun Jensen,[1] Victoria Sedor,[2] Emmanuel Eshun,[2] Patrick Njage,[1] Saria Otani,[1] Frank M. Aarestrup[1]

**ABSTRACT**  There is limited knowledge on the bacteriome and resistome in livestock in Africa and the potential influence of the animal husbandry practices and scale has also been scantly explored. We quantified and characterized the antimicrobial resistance gene (ARG) pools (resistomes) and bacteriome in 30 pigs and 60 poultry samples (free-range: rural and urban, and industrialized) across Ghana using Resfinder and Silva databases and compared them to similar data from pigs and poultry from nine European countries. The pig and poultry were very different in ARG and bacterial abundance and composition. The bacterial communities in the Ghanaian samples also differed substantially from the European samples, especially driven by a much higher abundance of *Subdoligranulum* in both animals. We found lower ARG loads in Ghanaian pigs compared to European counterparts. Among poultry, urban free-range samples exhibited lower ARG abundances compared to the lowest European levels, while rural free-range samples were comparable to the European average, and industrialized samples showed higher ARG abundances. Despite major differences in abundance, the resistome composition of Ghanaian urban, rural, and industrialized poultry samples was similar. Contrasts with European samples were mainly driven by increased abundances of different tetracycline resistance genes in Ghanaian samples, and increased abundances of ARGs encoding resistance to macrolides, beta-lactams, and trimethoprim in Europe. Among pig samples, the main differences were caused by ARGs encoding resistance to nitroimidazoles, beta-lactams, and macrolides in European samples. This study is, to the best of our knowledge, the first report on the resistome measured using metagenomics in livestock from Sub-Saharan Africa.

**IMPORTANCE**  To the best of our knowledge, this is the first report on the resistomes that are measured using metagenomics in livestock from Sub-Saharan Africa. We find notable differences in the microbiomes between both pigs and poultry, and those also varied markedly compared to similar samples from Europe. However, for both animal species, the same bacterial taxa drove such differences. In pigs and urban free-range poultry, we find a very low abundance of antimicrobial resistance genes (ARGs), whereas rural free-range poultry displayed similarity to the European average, and industrialized poultry exhibited higher levels. These findings show how different African livestock bacterial communities and resistomes are from their European counterparts. They also underscore the importance of continued surveillance and investigation into antimicrobial resistance across diverse ecosystems, contributing significantly to global efforts toward combating the threat of antibiotic resistance.

**KEYWORDS**  livestock, resistome, bacteriome, Africa, antimicrobial resistance genes

Antimicrobial resistance (AMR) is considered one of the largest threats to human and animal health (1, 2), and many studies have shown that AMR bacteria, and in some cases, antimicrobial resistance genes (ARGs) can transfer between livestock and humans (3, 4). ARGs are not necessarily confined to single bacterial species but may transmit between multiple species. Most studies have, however, focused on single species based

Address correspondence to Frank M. Aarestrup, fmaa@dtu.dk.

The authors declare no conflict of interest.

See the funding table on p. 12.

on laboratory cultivation and may have missed the emergence and dispersal of ARGs in the normal gut microbiota.

With the developments in sequencing technologies, it has become technically and economically feasible to characterize any animal microbiome and their associated ARG reservoirs, the resistome. It has also been documented that metagenomic methods offer improvements over phenotypic AMR surveillance (5, 6). There is, however, a sparsity of data, especially from Sub-Saharan Africa.

Major changes are currently happening in livestock production across Africa, with more industrialized livestock production existing alongside both urban and rural open-range production of pigs and poultry (7). This is also the case in Ghana where livestock production is increasing and changing from traditional small-scale farming towards an increasing production in large-scale farming (8).

There is, however, only limited information about the occurrence of AMR and especially a lack of data comparing different production systems. Thus, to the best of our knowledge, no study has as of yet been conducted comparing the resistome of rural and urban pigs and poultry and those under more industrial productions in Africa.

This study compares the resistome of fecal samples collected from free-range chickens in rural and urban settings, as well as chickens from industrial production. In addition, fecal samples from rural and urban pigs, as well as pigs produced under local biosecurity conditions, were also investigated.

## MATERIALS AND METHODS

### Pig samples

From February to August 2020, a total of 30 pig fecal samples were collected from semi-intensive and intensive systems in locations shown in Fig. S1. In the intensive system, pigs are housed without access to the outside environment, building structures are normally blocks or bricks, and the pen floors are cemented (Table S1). The roofing is either aluminum sheets or thatch. Animals are fed with home-prepared or commercial pig feed and water provided by owners, and herd size is normally above 50. Most exteriors are covered with fly netting. Cleaning is done at least once a day but normally workers clean twice a day. Animal health is provided by qualified veterinary personnel. Although there are instances where farm workers/owners administer medications. Large white and landrace are normally the breed of choice. In the semi-intensive system, pigs are provided with shelter and feed, although they get access to the outside environment to scavenge. Building structures are usually wooden or mud. The floor may be cemented; however, wooden floors are common and biosecurity measures are normally minimal or non-existent. Feeding and cleaning are not regular. Herd size is normally below 50 and animal health care is very poor. Farmers usually resort to self-medication. The breeds used are normally crossbreds (exotic). Pig feces were collected at 16 intensive and 14 semi-intensive farms across Ghana (Fig. S1; Table S1). Sampling was done early morning immediately following the cleaning and subsequent feeding of the pigs. From each farm, 10 fresh undisturbed floor fecal samples were collected from healthy pigs and later pooled into a single sample in the laboratory. The samples were stored on ice during sampling and transported to the National Food Safety Laboratory within 24 hours.

### Poultry samples

A total of 60 chicken caecal samples were collected in the Greater Accra region between January and May 2020. Healthy-appearing chickens were purchased from live bird markets within Accra (Ashaiman market, Kaneshie market, Dome market), one rural market (Ada), and one urban community (Weija area) where there is no wet market (Table S2).

All birds purchased from the Weija community were transported in cages to Kaneshie (the closest live bird market), where they were slaughtered. The rural free-range chickens

were purchased from Ashaiman and Ada live bird markets. Slaughtered birds arrive in central markets via intermediaries, who typically transport them from rural farms. The urban free-range chickens were purchased from three different locations: Kaneshie live bird market, Dome live bird market, and Weija community within Accra.

The industrial chickens were purchased from Kaneshie, Dome, and Ada live bird markets. These birds have spent layers from big commercial farmers clearing their old stocks in preparation for new birds. Approximately, 10 samples were collected every 2 weeks, with some interruptions due to the COVID-19 lockdown.

After slaughtering, the whole intestine was removed and placed on a decontaminated plastic sheet (with 70% EtOH) laid on a bench to create a working bench. The intact caecum of each bird was aseptically extracted with sterile forceps and scissors, placed in a whirl Pac or sterile falcon tubes labeled with the sample ID, and immediately stored in a cooler box with ice packs before transporting to the National Food Safety Laboratory on the same day. The caecum content was extracted and shipped frozen to Technical University of Denmark (DTU) for DNA extraction and sequencing.

## DNA purification and sequencing

Total DNA was purified from all caecal and fecal samples using the QIAamp Fast DNA stool mini kit (Qiagen, Germany) following the manufacturer's instructions using 200 mg as starting material. The DNA was eluted in 50 µL of pre-heated (65°C) AE buffer to increase DNA yield. DNA quality was checked using a Qubit Fluorometer (Thermo Fisher Scientific). For metagenomics sequencing, all libraries were prepared using the PCR-free Kapa Hyper Prep Kit (Roche). All libraries were sequenced on Illumina Novaseq 6000 S4 (2 × 150 bp) platform.

## Preprocessing and mapping of sequencing reads

The raw sequence reads were quality checked (FastQC version 0.11.5 https://www.bioinformatics.babraham.ac.uk/projects/fastqc/) and trimmed (BBduk2 version 36.49) (9) to remove adaptors and low-quality sequences. Read assignments to reference databases were performed with KMA 1.3.27 (10). ResFinder database (2020–01-25) (11) and Silva 16S rRNA database (2020–01-16) were used to assign ARGs and bacterial taxa, respectively, in the metagenomic samples. To compare our pig and poultry microbiomes to other data sets, we have included previously sequenced pig and poultry microbiomes data from 181 pigs and 178 chickens from nine countries in Europe (Belgium, Bulgaria, Germany, Denmark, Spain, France, Italy, the Netherlands, and Poland) (12). Those additional microbiomes are part of the EFFORT project (http://www.effort-against-amr.eu/) and are available from the European Nucleotide Archive (ENA) with the project accession number: PRJEB22062. In short, both the pig and poultry samples were collected from conventional farms with an all-in-all-out production.

## Antimicrobial resistance gene quantification

Total antimicrobial resistance per sample was quantified by calculating the total AMR fragments per kilobase per million fragments per sample (FPKM, equation (1)) and visualized in boxplots.

$$\text{Relative abundance} = \frac{\text{ARG}_{\text{Fragments}}}{\text{ARG}_{\text{Length}} \cdot \text{Bacteria}_{\text{Depth}}} \cdot 10^9 \qquad (1)$$

Here, $\text{ARG}_{\text{Fragments}}$ is the number of fragments assigned to a reference sequence, $\text{ARG}_{\text{Length}}$ is the ARG reference length, and $\text{Bacteria}_{\text{Depth}}$ is the sum of fragments assigned to superkingdom Bacteria determined by Silva.

The relative abundance was also calculated for the difference resistance gene classes (classification scheme as given by the official ResFinder database documentation: https://bitbucket.org/genomicepidemiology/resfinder_db/src/master/) and summarized in a stacked barplot to investigate the composition of each sample.

## Compositional data analysis

The read fragment counts were used as the gene and taxa counts for mapping against the ResFinder (2020–01-25) and Silva (2020–01-16) databases. Only fragments assigned to superkingdom Bacteria were used for the Silva mapping. The counts were normalized by the reference length divided by 1,000 to avoid small numbers which might interfere with the downstream zero replacement.

To account for the compositional nature of the microbiome data sets (13), zero replacement on the length-adjusted counts was performed followed by a centered-log ratio (CLR, equation (2)) transformation:

$$x = [x_1, \ldots, x_D] \tag{2}$$
$$clr(x_1, \ldots, x_D) = \left(\log\left(\frac{x_1}{G(x)}\right), \ldots, \log\left(\frac{x_D}{G(x)}\right)\right),$$
$$where\ G(x) = \sqrt[D]{x_1 \cdot, \ldots, \cdot x_D}$$

To select the features, i.e. number of features should be less than the number of samples, only the most variant, most abundant features were included for the PCA.

## Statistical analyses

The differential abundance analysis was performed with ALDEx2 version 1.18.0 (14) in R. Differences in the centered-log ratio (CLR) abundance between groups was tested with a Welch's $t$-test followed by a Benjamini-Hochberg false-discovery rate (FDR) correction (15). The CLR variation within-group as well as between-group was shown in an effect plot, highlighting the statistical significant resistance genes with FDR < 0.05 in red. The effect of these statistically significant resistance genes was reported for each group. All features were used for this analysis with no filtering applied.

## Diversity measures

Richness, diversity, and evenness diversity measures were reported for the length-adjusted fragment counts. Shannons' evenness, chao1, and Simpson diversity index were all calculated using the skbio.math.diversity.alpha python package (http://scikit-bio.org/docs/0.1.3/math.diversity.alpha.html#module-skbio.math.diversity.alpha)

## Data visualization

Boxplots and stacked barplots were visualized with Python 3.8, Matplotlib 3.7.1 (16), and Seaborn 0.12.2 (17). The ordination analysis and the PCA visualizations were performed and created with the Python pyCoDaMath package (https://pypi.org/project/pyCoDa-Math/). An additional function was created within the package to make the centroid sample point lines of the PCA figures. Barplots visualizing the differential abundance analysis results were made with the ggplot2 library (18) while the differential abundance analysis was performed with ALDEx2 version 1.18.0 (14) in R version 3.6.3.

## RESULTS

### Summary of the data

A total of 5.48 billion PE reads were obtained from the 90 samples (average 60.9 million PE reads per sample, range 15.7–154.3 million PE reads per sample, SD: 189 million) (Fig. S2; Table S3). On average, 0.119% of the reads per sample aligned to ARGs from the ResFinder database, and 0.291% of the reads per sample aligned to 16/18 S SSU rRNA from the Silva database. Of which, an average of 96.199% per sample was assigned to Bacteria superkingdom and 3.674% per sample to eukaryotes. An average of 316 unique bacterial genera were detected per sample (range 146–537 bacterial genera, SD: 60)

(on average, 6,243 genome equivalents identified with MicrobeCensus per sample). The most abundant bacterial genera (highest centered log-ratio median across samples) in pig feces were *Subdoligranulum*, *Streptococcus,* and *Lactobacillus* and *Subdoligranulum*, *Streptococcus,* and *Olsenella* in chicken feces (Table S4).

## The acquired resistome

A total of 827 different ARGs were observed across the 90 samples from Ghana; 544 among poultry and 688 among pigs. The most abundant ARGs (highest centered log-ratio median across the samples) in the poultry feces were *tet*(W), *tet*(Q), and *aph*(3')-III. In the pig feces, the most abundant ARGs were *tet*(W), *ant*(6)-Ia, and *tet*(O/W) (Table S5).

On average, ARG abundances were lower among both urban and rural pigs from Ghana compared to any of the nine countries in Europe that were included here for comparison. Abundances in ARGs from urban poultry in Ghana were lower than the majority of the ARGs observed in European poultry (Fig. 1A). By contrast, the ARGs from rural poultry were on the same average as European poultry, while industrialized poultry from Ghana showed higher abundances than those from any country in Europe.

The number of reads assigned to different ARGs was summed for each antimicrobial drug class (Fig. 1B). When examining the relative distribution of ARGs encoding resistance to different antimicrobial classes, there was in general a lower abundance of resistance to beta-lactams and folate pathways among the samples from poultry in Ghana compared to Europe. Among urban poultry, there was a relatively higher abundance of ARGs encoding resistance to other antimicrobials and macrolides. Among pig samples, the most evident was a relatively lower abundance of resistance to

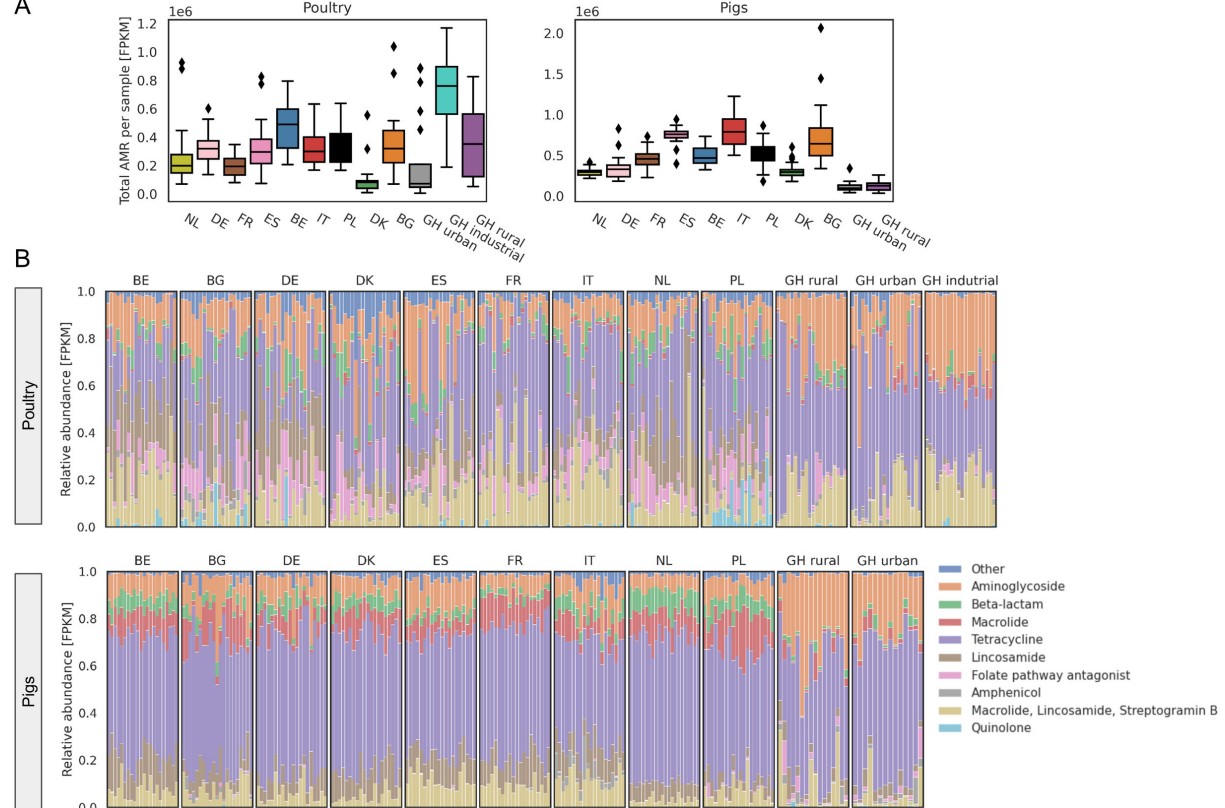

**FIG 1** Overview of AMR abundance and composition of pig and poultry from Ghana and nine European countries. (A) Total AMR level per sample per country in poultry (left) and pigs (right). (B) AMR gene class composition per sample per country, stratified by host. The two-letter country codes are used in all figures: BE, Belgium; BG, Bulgaria; DE, Germany; DK, Denmark; ES, Spain; FR, France; GH, Ghana; IT, Italy; NL, the Netherlands; PL, Poland.

macrolides and a higher abundance of resistance to aminoglycosides when comparing Ghana to Europe.

When comparing the resistome composition of poultry and pig samples individually, samples from Ghana were different from European samples (Fig. 2A and D). Despite the major differences in abundance, this was also the case for urban, rural, and industrialized poultry samples that all clustered together and separately from European samples. These differences were mainly driven by increased abundances of different tetracycline resistance genes among the poultry samples from Ghana and increased abundances of ARGs encoding resistance to macrolides, beta-lactams, and trimethoprim among samples from Europe (Fig. 2C; Table S7). Among pig samples, the main differences were mainly caused by ARGs encoding resistance to nitroimidazoles, beta-lactams, and macrolides in European samples (Fig. 2F; Table S6).

The investigation focused only on the samples from Ghana showed a clear host species effect on resistance gene clustering, with 46.5% variance explained by the first of two components (Fig. 3A). The separation was mainly caused by higher abundances of different tetracycline resistance genes among the poultry samples, noticeably an overrepresentation of *tetA*(P) and *tetB*(P), as well as *bla*ACL and *str* (Fig. 3D; Table S8). In the pig samples, amphenicol resistance was driving the separation, mainly *cat_3* and *cfr*(C), as well as the presence of *tet*(X) (Table S8). The host species effect was also

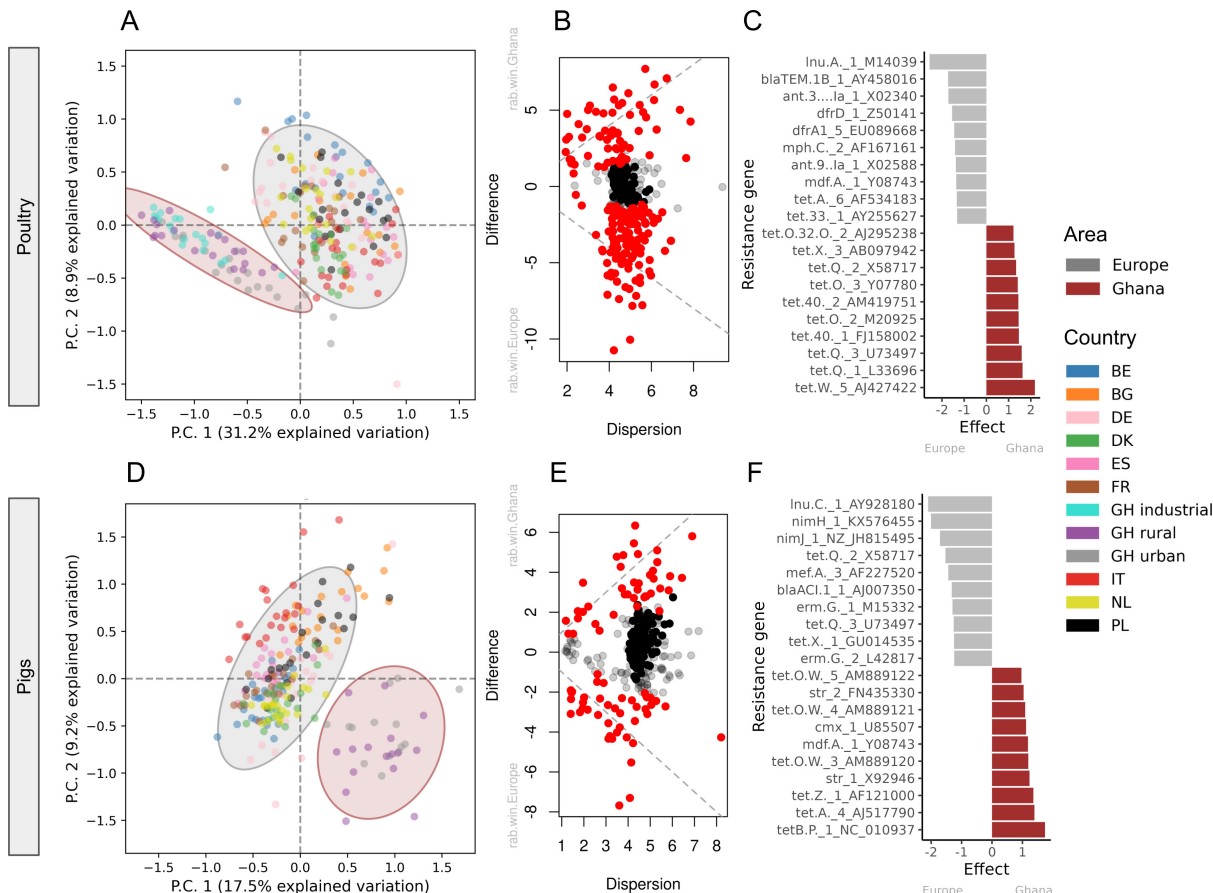

**FIG 2** Differences in resistance genes between pig (D–F) and poultry (A–C) from Ghana and nine European countries. (A) Poultry resistance genes clustering. The ordination analysis was performed on the most abundant, most variant-centered log-ratio (CLR) transformed size-adjusted counts (CLR variance >1, CLR median >0). (B) Poultry effect plot from the differential abundance analysis is further investigated in (C). Top 10 statistically significant poultry resistance genes between Europe and Ghana with FDR correction <0.05 (see Table S7 for full list). (D) Pig resistance genes clustering. The ordination analysis was performed on the most abundant, most variant CLR transformed size-adjusted counts (CLR variance >1, CLR median >0). (E) Pig effect plot from the differential abundance analysis is further investigated in (F). Top 10 statistically significant pig resistance genes with FDR correction <0.05 (see Table S6 for full list).

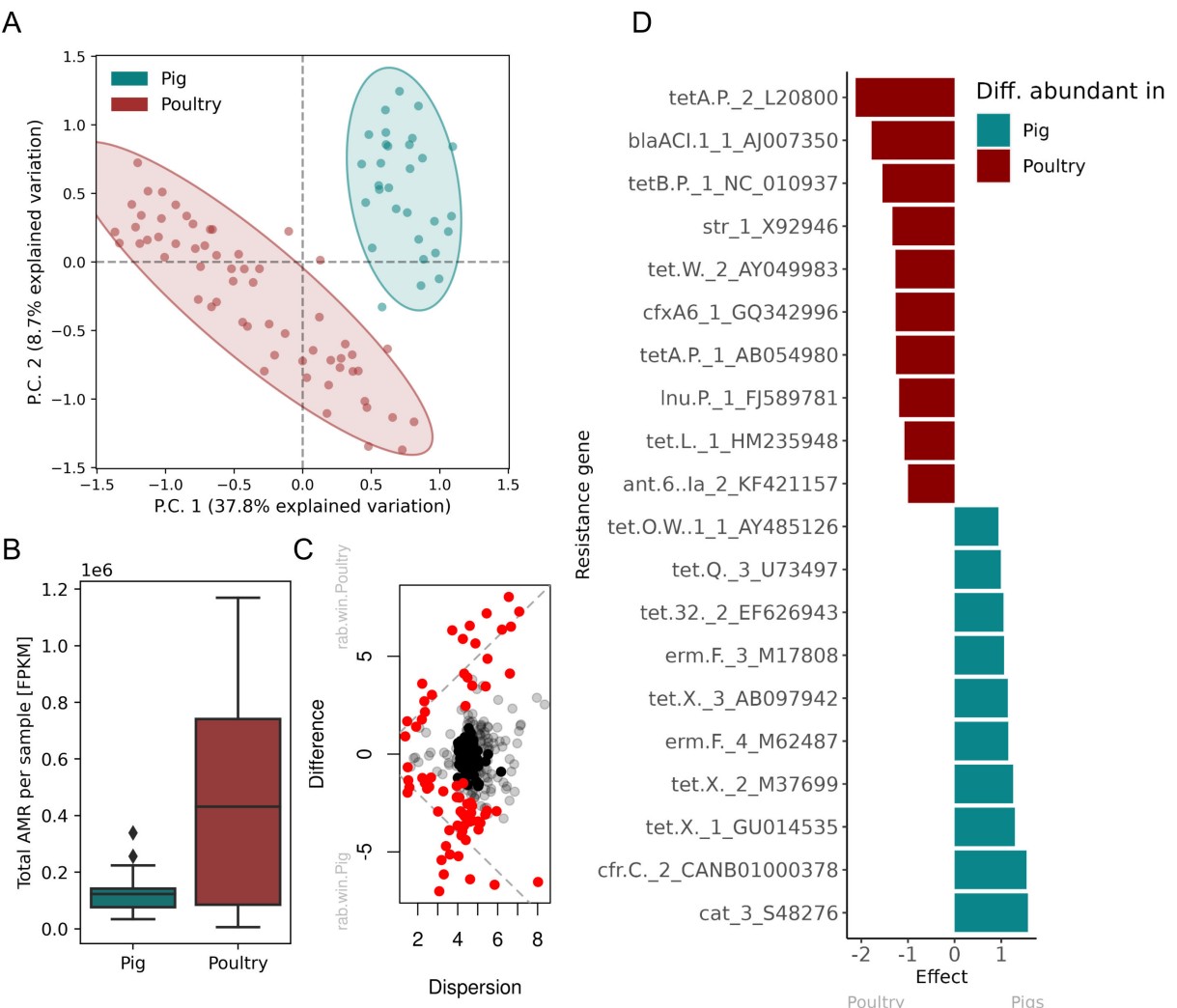

**FIG 3** Differences in resistomes of pig and poultry from Ghana. (A) Resistance genes PCA clustering. The ordination analysis was performed on the most abundant, most variant-centered log-ratio (CLR) transformed size-adjusted counts (CLR variance >2.2, CLR median >0). (B) Total AMR per sample is calculated as the total AMR fragments per kilobase per million fragments per sample (FPKM), stratified by the host. (C) Differential abundance effect plot showing the within-group dispersion of CLR values of each resistance gene compared to the between-group differences. Statistically significant resistance genes with a Benjamini-Hochberg false-discovery rate (FDR) correction <0.05 are colored red. The gray dotted line indicates an effect size of 1. (D) Top 10 statistical significant resistance with FDR correction <0.05 identified from the differential abundance analysis (see Table S8 for full list).

reflected by the AMR levels. Ghanaian pigs and poultry differed significantly in their AMR level, with poultry having a higher median as well as a wider spread (Fig. 3B).

## Resistome variation within hosts

While the samples from urban or rural pigs showed no difference in clustering even with the more detailed investigation (Fig. S3F through K), differences between industrial and urban poultry were observed (Fig. S3D). The differential abundance analysis showed that this separation was mainly driven by *ant*(6)-Ia, *aph*(3′)-III, *tet*(W), and *cfr*(C) (Fig. S3E) present in industrial poultry. This difference was also observed in the AMR load differences in rural, urban, and industrial poultry (Fig. S4A). Here, industrial poultry had a significantly higher AMR level than both rural and urban poultry, with urban poultry having the lowest.

No difference in AMR load between rural and urban pigs was observed (Fig. S4B) and increased biosecurity in itself did not influence this load (Fig. S4C). Some difference

in AMR load was however observed between urban and rural pigs with increased biosecurity, with the rural high-end having a higher AMR load than the urban high-end (Fig. S4D). However, since the sample size is quite small, this would need further investigation.

## Bacterial community

Ghanaian poultry microbiomes were dominated by *Subdoligranulum, Streptococcus, Olsenella, Bacteroides,* and *Myroides*, while *Subdoligranulum, Streptococcus, Lactobacillus, Bifidobacterium,* and *Myroides* dominated pig microbiomes. Poultry and pig microbiomes varied markedly using our ordination analyses when comparing their bacterial taxa (Fig. 4A) with 157 bacteria that contribute to this host-bacterial specificity (Table S9). Of these, *Streptococcus, Paeniclostridium, Clostridioides,* and *Lactobacillus* were the most differential abundant in pig feces, and *Alistripes, Mordavella, Bacteroides,* and *Enorma* in poultry feces (top 10 most differential abundant bacterial genera summarized in Fig. 4C. Refer to Table S9 for all differential abundant bacterial genera).

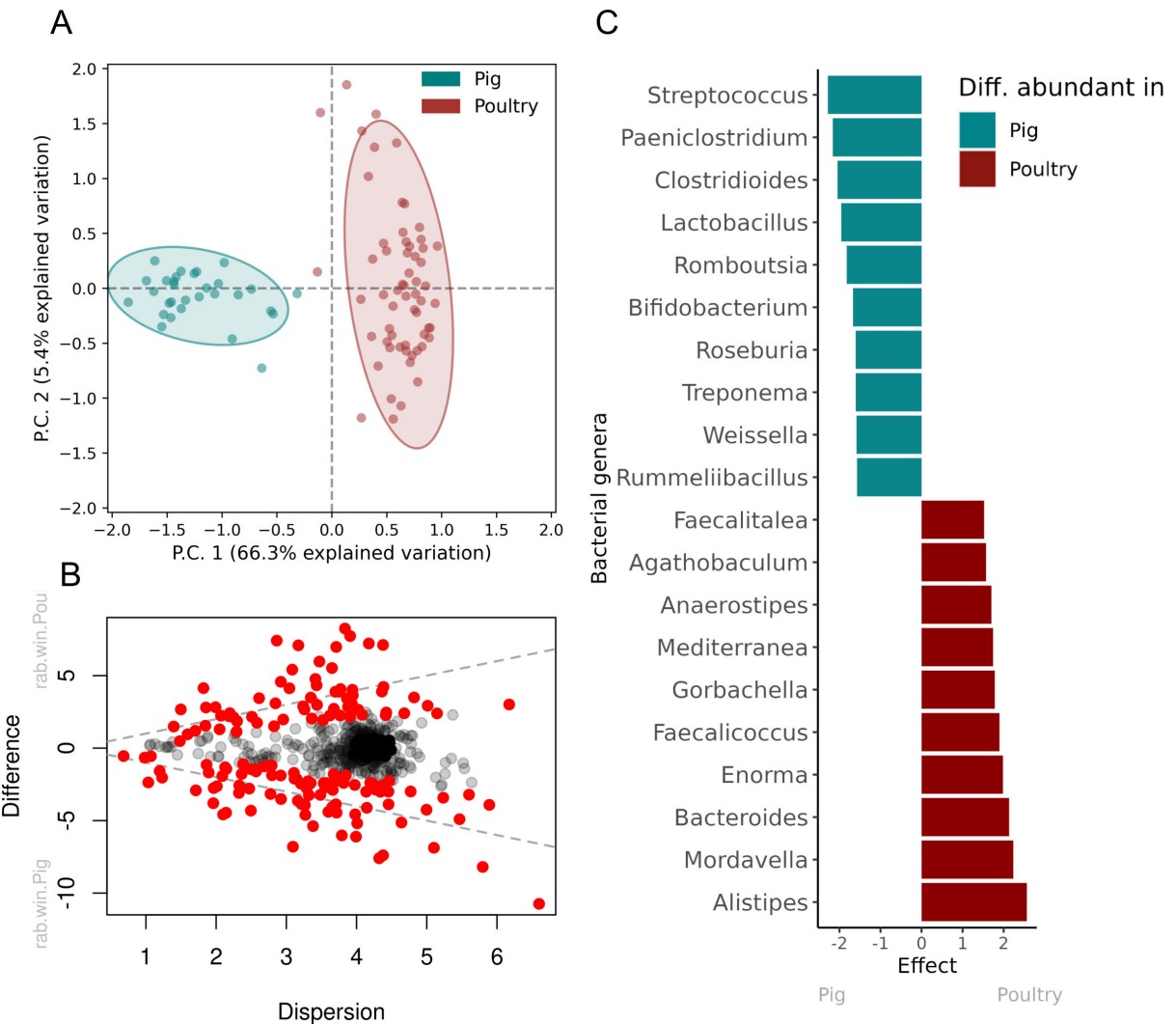

**FIG 4** Bacterial genus differences between pig and poultry in Ghana. (A) Clustering of pig and poultry samples from Ghana. The ordination analysis was performed on the most abundant, most variance centered log-ratio (CLR) transformed size-adjusted counts CLR variance >2, CLR median >0). (B) Effect plot showing the within-group dispersion of CLR values of each bacterial genus compared to the between-group differences. Statistical significant bacterial genera with a Benjamini-Hochberg false-discovery rate (FDR) correction <0.05 are colored red. The gray dotted line indicates an effect size of 1. (C) Top 10 statistically significant bacterial genera with FDR correction <0.05 (see Table S11 for full list).

There were no differently abundant bacterial taxa that separate bacterial communities within a host, for example, no bacteria were significantly different between urban and rural pig microbiomes (Fig. S5).

We also observed a clear separation in the bacterial communities between Ghanaian poultry and pig feces and the European ones (Fig. 5 A and D), with the first two components explaining more than 50% of the variation. In poultry, this variation was explained by 203 differential abundant bacterial genera (Table S10) and in pigs, it was explained by 138 differential abundant bacterial genera (Table S11). Interestingly, increased abundances of *Subdoligranulum* in samples from Ghana were the main driver for both pigs and poultry (Fig. 5C and F). Other genera also separated the African pigs and chickens from their European counterpart microbiomes; for example, *Collinsella* and *Olsenella*, and *Bifidobacteria* and *Clostridium*, in chicken and pigs, respectively.

## Alpha diversity and richness

We calculated several alpha-diversity indices (Chao1, Simpsons and Shannon's evenness) both in the microbiome (Fig. S6) as well as within the resistome (Fig. S7). In general, pigs had higher diversity and richness of bacterial genera than the poultry samples. No significant difference in average alpha diversity was observed within poultry samples (rural, urban, and industrial). For ARGs, a similar tendency was observed and also the diversity of ARGs among industrialized poultry was slightly higher.

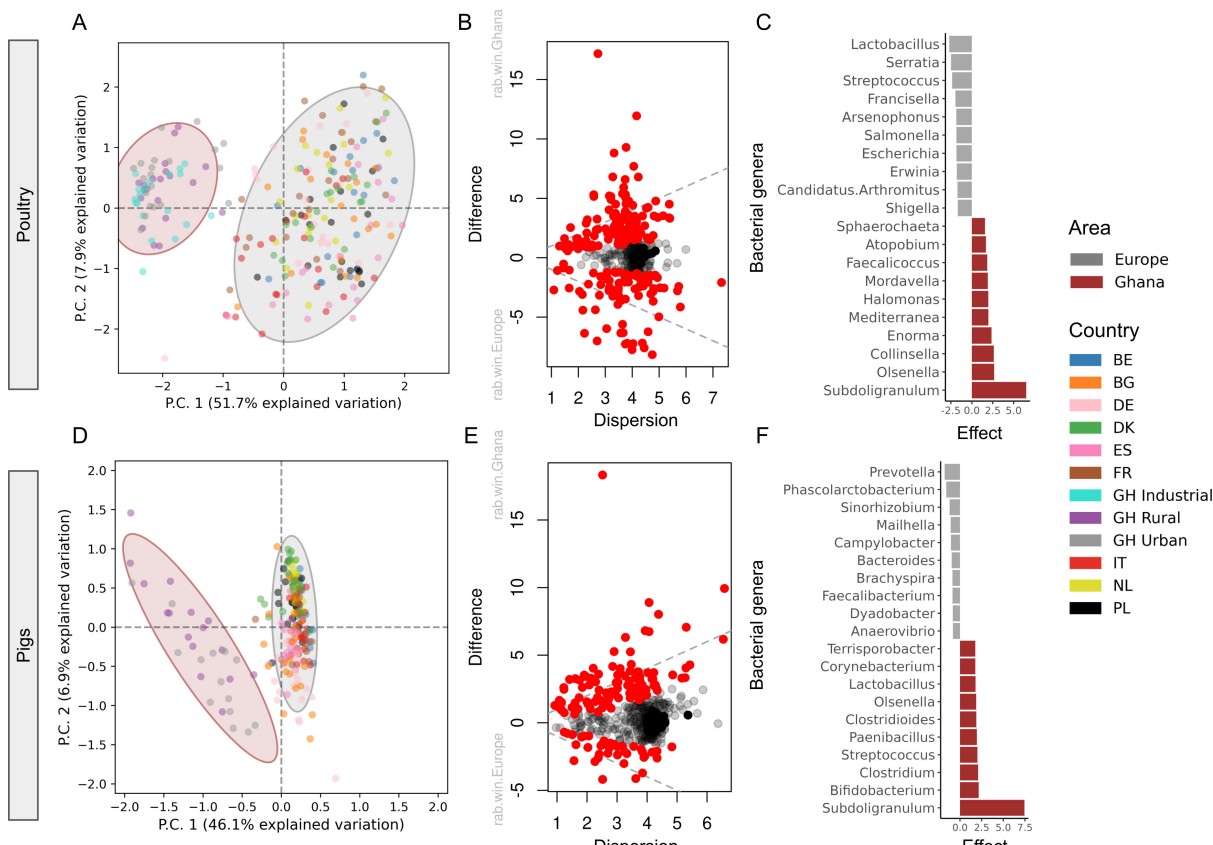

**FIG 5** Differences in bacteria genera between poultry (A–C) and pigs (D–F) from Ghana and nine European countries. (A and D) Bacteria genera clustering. The ordination analysis was performed on the most abundant, most variant centered log-ratio (CLR) transformed size-adjusted counts (CLR variance >1, CLR median >0.5). (B and E) Effect plot showing the within-group dispersion of CLR values of each resistance gene compared to the between-group differences. Statistical significant genera with a Benjamini-Hochberg false-discovery rate (FDR) correction <0.05 are colored red. The gray dotted line indicates an effect size of 1. (C and F) Top 10 statistically significant genera with FDR correction <0.05 (see Table S13 for full list).

## DISCUSSION

There is a scarcity of data on AMR from livestock in Africa (19) and often data are difficult to compare because studies report resistance among different bacterial species, different isolation, and characterization methods, or toward different antimicrobial agents. To the best of our knowledge, there is no other report on the resistome of pigs and poultry from Ghana or from Sub-Saharan Africa, reporting abundances of ARGs.

Interestingly, we observed a lower abundance of ARGs among pigs and urban free-range chickens from Ghana compared to data from Europe. However, for industrialized chickens, the ARG abundances were higher in Ghana compared to any country in Europe. This is in agreement with previous studies also showing lower levels of AMR among small-scale and free-range production compared to more industrialized production systems (20–22), and confirms predictions that the transition from small-scale and free-range production to large-scale industrialized production will be accompanied with an increase in AMR (23). Poultry samples from rural free-range poultry showed abundances similar to the European average. This is in contrast to what we would have expected, namely that urban free-range would be associated with more ARG dispersal from humans and thus higher ARG loads compared to rural free-range. There are unfortunately no other similar studies to compare to, but this is an interesting observation and should be further investigated. It should also be mentioned that industrialized chicken samples were mainly from older layer chickens, where there might have been additional time to build up increased resistance abundances.

There is a paucity of antimicrobial consumption data for livestock in Africa, but according to data from WOAH, Africa seemingly has the lowest usage of antimicrobials in animals in the world, when adjusted by animal biomass (24). This might explain the observed very low abundance of ARGs among the samples. However, without antimicrobial usage data from livestock production in Ghana, this is difficult to tell. Other studies using phenotypic susceptibility testing of bacteria isolated from pigs and poultry in Ghana have found relatively high levels of resistance (25–27), which is in some contrast to our findings. However, we would also like to mention that for some of these phenotypic studies, spurious resistance patterns are reported, potentially suggesting problems with the tests. It is not the purpose of this study to review previously published studies, but we would like to mention that one of the advantages of using next-generation sequencing including for metagenomics is the possibility to share and re-analyze the raw data, which is not feasible for reported phenotypic summary results.

Previous studies from other regions of the world have shown a major effect of the animal host and the country of origin on the resistome (12, 28–32). Here we compared the resistome data from Ghana to previously published data from Europe (12). We restricted the comparison to these data since they were generated using the same DNA-purification methodology and thus limiting bias (33–35).

We observed a major separation according to the animal host, however, importantly also a very separate resistome of both pigs and poultry from Ghana compared to those observed from Europe. Thus, in addition to a major impact of the host on the resistome, country-specific factors clearly also impact the resistome. Similar observations have been made in other studies (12, 29–31). No differences in the resistome of the different groups of pigs were observed, but interestingly we observed a separation of urban poultry from the two other groups. The most important ARGs were *aph*(3′)-III, *ant*(6)-Ia, *cfr*(C), and *tet*(X) which were observed in higher abundances in the latter groups. *aph*(3′)-III and *ant*(6)-Ia both confer resistance to aminoglycosides and are very frequently found as a part of the core resistome in many animal species. The frequent occurrence of *cfr*(C) and *tet*(X) is difficult to explain.

In a metagenomic study of all available shotgun metagenomic sequencing data from NCBI SRA, Lawther et al. (36) reported tetracycline resistance as being the most widespread resistance occurring in both pig and poultry microbiomes (limited to countries from Europe, Asia, and North America). This is in line with our findings, with *tet*(W) being the most abundant resistance gene in both Ghanaian pig and poultry

microbiomes (Table S2). Even with ARG levels being lower in Ghana compared to the nine European countries compared in this study, global trends are reflected in the resistomes of Ghanaian pigs and poultry.

Previous studies showed that host resistomes could be shaped by their bacterial communities (12, 37). A number of bacterial taxa shaped the Ghanaian pig and poultry microbiomes differently from their European counterparts (Fig. 5). Such variations in those microbiomes are likely to be stable due to the bacterial functions that are provided to the host. For example, *Subdoligranulum* was the main bacterium that signified the Ghanaian pig and poultry microbiomes. It is known to produce butyrate and has previously been known for its role in fermenting dietary fibers (38). This might indicate a higher fiber diet in these animals relative to their European counterparts. Species belonging to the genera *Collinsella* and *Olsenella* have previously been identified from chickens (39, 40) and *Collinsella* has been previously linked to obesity, atherosclerosis, pro-inflammatory dysbiosis, and inflammatory burden in human studies (41–44). The potential importance of these bacterial genera in poultry from Ghana warrants further studies. Among Ghanaian pigs, a higher abundance of *Bifidobacterium* and *Clostridium* was observed than among their European counterparts. *Bifidobacterium* are some of the first microbes to colonize the human gastrointestinal tract, whereas *Clostridium* are more common in older humans; they are associated with several positive health benefits and increased abundances are observed when on a fiber-rich diet (45–48).

Biosecurity has also been highlighted as a potentially important factor for reducing AMR in livestock farming, either through reducing diseases and thereby AMU, or directly by restricting transmission of AMR (49). There are, however, few studies that have investigated this. In previous studies examining the importance of biosecurity on the resistome in poultry and pig farming in Europe, it was not possible to identify an association for poultry production, whereas in pig farming increased biosecurity was associated with increased AMR (50, 51). In this study, we did not find any effect of biosecurity in pig farming on the resistome. This might be because of the limited number of pigs sampled, but suggest that it is not a main factor influencing the pig resistome.

In conclusion, this study is to the best of our knowledge the first report on the resistome measured using metagenomics in livestock from Sub-Saharan Africa. We find a very different microbiome among both pigs and poultry compared to similar samples from Europe. For pigs and urban free-range poultry, we find a very low abundance of ARGs, whereas abundance in rural free-range poultry is similar to the European average, and abundance in industrialized poultry is higher in Ghana.

## ACKNOWLEDGMENTS

We are grateful to Birthe S. Rosenqvist Lund, Hanne Mordhorst, and Baptiste Jacques Philippe Avot for technical assistance.

This study was funded in part by the Novo Nordisk Foundation (Grant: NNF16OC0021856: Global Surveillance of Antimicrobial Resistance) and the Fleming Fund through fellowships to V.S. and E.E.

V.S., E.E., P.N., and F.M.A. conceptualized and designed the study. V.S. and E.E. performed sampling and collection of epidemiological data. F.M.A. obtained funding, P.N. organized shipment, and S.O. performed sequencing. E.E.B.J. performed the bioinformatics and statistical analyses and made all figures. F.M.A. drafted the manuscript together with all other authors. All authors have seen and approved the final manuscript.

## AUTHOR AFFILIATIONS

[1]Technical University of Denmark, Kemitorvet, Denmark
[2]Veterinary Services Department, Ministry of Food and Agriculture, National Food Safety Laboratory, Accra, Ghana

## AUTHOR ORCIDs

Emilie Egholm Bruun Jensen  http://orcid.org/0000-0002-3214-7918
Saria Otani  http://orcid.org/0000-0002-2538-8086
Frank M. Aarestrup  http://orcid.org/0000-0002-7116-2723

## FUNDING

| Funder | Grant(s) | Author(s) |
|---|---|---|
| Novo Nordisk Foundation | NNF16OC0021856: Global Surveillance of Antimicrobial Resistance | Victoria Sedor |
|  |  | Emilie Egholm Bruun Jensen |
| The Fleming Fund | Fellowships | Victoria Sedor |
|  |  | Emilie Egholm Bruun Jensen |

## AUTHOR CONTRIBUTIONS

Emilie Egholm Bruun Jensen, Data curation, Formal analysis, Software, Visualization, Writing – original draft | Victoria Sedor, Conceptualization, Funding acquisition, Investigation, Project administration, Resources, Supervision, Writing – original draft, Writing – review and editing | Emmanuel Eshun, Conceptualization, Formal analysis, Investigation, Methodology, Resources, Writing – review and editing | Patrick Njage, Conceptualization, Supervision, Writing – review and editing | Saria Otani, Conceptualization, Formal analysis, Investigation, Methodology, Resources, Writing – review and editing | Frank M. Aarestrup, Conceptualization, Funding acquisition, Investigation, Project administration, Resources, Supervision, Writing – original draft, Writing – review and editing

## DATA AVAILABILITY

The metagenomic sequencing data (FASTQ) from the 60 poultry and 30 pig feces samples have been deposited and can be accessed through the European Nucleotide Archive (ENA) with the following project accession number: PRJEB62878.

## ADDITIONAL FILES

The following material is available online.

### Supplemental Material

**Supplemental tables and figures (mSystems00629-23-s0001.docx).** Tables S1 to S11 and Figures S1 to S7.

### Open Peer Review

**PEER REVIEW HISTORY (review-history.pdf).** An accounting of the reviewer comments and feedback.

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
