## [Reviewer comments · mSystems]

The resistomes of rural and urban pigs and poultry in Ghana

Emilie Jensen, Victoria Sedor, Emmanuel Eshun, Patrick Njage, Saria Otani, and Frank Aarestrup

Corresponding Author(s): Frank Aarestrup, Danmarks Tekniske Universitet

Review Timeline:

Submission Date:	June 15, 2023
Editorial Decision:	July 18, 2023
Revision Received:	August 1, 2023
Accepted:	August 2, 2023

Editor: Rachel Poretsky

Reviewer(s): The reviewers have opted to remain anonymous.

Transaction Report:

DOI: <https://doi.org/10.1128/msystems.00629-23>

July 18, 2023

Prof. Frank M. Aarestrup
Danmarks Tekniske Universitet
National Food Institute
Kemitorvet
Lyngby 2800
Denmark

Re: mSystems00629-23 (The resistome of rural and urban pigs and poultry in Ghana)

Dear Prof. Frank M. Aarestrup:

Thank you for submitting your manuscript to mSystems. We have completed our review and I am pleased to inform you that, in principle, we expect to accept it for publication in mSystems. However, acceptance will not be final until you have adequately addressed the reviewer comments. That said, the reviewers were both positive and I agree that this study adds information to a region of the world where such data is lacking. Additional context about antimicrobial use in animals (and humans) in Ghana would be a helpful addition.

Preparing Revision Guidelines

Please return the manuscript within 60 days; if you cannot complete the modification within this time period, please contact me. If you do not wish to modify the manuscript and prefer to submit it to another journal, please notify me of your decision immediately so that the manuscript may be formally withdrawn from consideration by mSystems.

Sincerely,

Rachel Poretzky

Editor, mSystems

Journals Department
Reviewer comments:

Reviewer #1 (Comments for the Author):

This work by EEB Jensen et al is an excellent manuscript that will be of great interest to scientists working with animal resistomes and microbiomes, as well as other researchers, including microbial ecologists and genomicists. I enjoyed reading it, and I believe the core technical analysis (on which all of the important biology in this type of manuscript rests) has been carefully designed and performed. Also, I have to congratulate the authors on the excellent and self-explanatory figures they designed, which unfortunately is not always the case. I have read the manuscript thoroughly and have only a few suggestions for the authors.

Main points:

It is unfortunate that there are no studies that the authors can use for relevant comparisons using poultry and pig samples from other countries besides Ghana and the 9 EU countries. It is certainly understandable that comparisons using metagenomic samples from different studies will introduce a significant bias rendering it impossible to draw any solid conclusions. However, maybe the authors could discuss a little more results from previous metagenomic studies in the resistome of poultry and pigs from other parts of the world, especially since Africa is considered a continent with the lowest antimicrobials usage in the world.

Throughout the figures and the manuscript text, there are interesting comparisons between the resistomes of the EU countries and the industrialized, rural, and urban resistomes of Ghana, but there is no info about the EU countries' resistomes. Were they collected from industrialized facilities or ecological farms? Considering the details about the Ghana sampling, the manuscript would benefit greatly if the authors provided some details about the origin of the EU samples. I understand these details may exist in the cited literature (and the effort website), but it would be nice if that information is also readily available in the manuscript. In short, this information should include the type of facilities (free range, industrialized etc), the type of feed (whenever possible), and the antibiotics usage (again, depending on the availability of information). If the metagenomic samples from the EU countries came from different types of facilities, maybe, it would make sense to separate and present these samples to industrialized/rural/urban, similarly to what was done for the Ghana samples.

Minor comments:

L79: do the authors have any details about the age of the animals?

L83: are any details available about the composition of the commercial feed and how it may differ from the European?

L89: do you know how frequently they were allowed to scavenge outside?

L92: what does "self-medication" entail?

L96: collected in sterile containers? falcon tubes?

L102: please remove "apparently"; either they were purchased from the markets or not

L107-109: sentence is a bit too colloquial, can you replace it with something like "Slaughtered birds arrive in central markets via intermediaries, who typically transport them from rural farms."

L115: please specify how long the interruptions were

L119: this is the first mention, please specify which nine countries

L159: please specify which versions or when were the databases downloaded

L207: I may have missed that, but how exactly did you estimate the number of genera per sample? Based on the 16S? Because, if yes, that is not ideal.... (I would suggest sth like MicrobeCensus for such estimations)

L410-411: My point again, regarding more information about the diets (whenever possible)

Reviewer #2 (Comments for the Author):

In their manuscript "The resistome of rural and urban pigs and poultry in Ghana", the authors determine metagenomic data in livestock from Sub-Saharan Africa with a focus on antimicrobial resistance genes and bacterial composition. The authors identify differences between livestock microbiota in Ghana and Europe, which contributes to a better understanding of the global microbiome diversity and warrants further investigation to complete a microbiota catalogue encompassing livestock species and breeds, countries, and different production systems. The data are solid, well-presented and support the conclusions drawn by the authors. I have only a few comments to the manuscript.

Major points

Results, lines 269, 270: tetA(P) and tetB(P) are frequently associated with Clostridium and blaACI also with anaerobic bacteria. Can the authors comment on this, as Clostridia were identified as being among the more abundant genera in the pig samples (Table S4)?

Discussion, Effects of host breed and diet on gut microbiome community and resistome: The bacterial composition and the resistomes differ strongly between pigs and poultry in Ghana and between the Ghanaian samples and the European samples. Differences in diet and in breeds present in the animal populations surely contribute the differences in microbiota and resistomes. Due to the means of poultry acquisition, there is no metadata as to antimicrobial usage and diet. This should be mentioned as a limitation to the study. Future studies should try to include such data.

Discussion: Was or will a metabolic reconstruction of the metagenome be attempted. This is beyond the scope of this study, but it would be interesting to compare metabolic profiles of European and Ghanaian samples to identify differences and try to backtrace dietary influences.

Minor points

Title: Since the authors stress that the pig and poultry metagenomes differ strongly in bacterial composition and in antibiotic resistance genes from each other, they might want to change the title to "The resistomes of rural and urban pigs and poultry in Ghana".

Introduction, line 56: change spacity to sparsity or to scarcity.

M&M, Pig samples, lines 84-87: change Netting to netting, Veterinary to veterinary and medications to medications.

M&M, Poultry samples: were the chickens from rural and urban sources also layers or broilers? Is there any information on the chicken breed or breeds that were slaughtered?

M&M, Antimicrobial resistance gene quantification, line 154: which scheme was used to classify the resistance genes?

Results, Summary of the data, line 203: the standard deviation of the sample is quite high, although there is only one sample each with a very low and one with a very high sample count. It might be easier to evaluate the homogeneity of the sample if a size distribution plot of the fragment counts were given in the supplemental material.

Results, line 267: change resistant to resistance.

Results, line 292 replace cfc(C) with cfr(C).

Response to reviewers

Reviewer #1 (Comments for the Author):

This work by EEB Jensen et al is an excellent manuscript that will be of great interest to scientists working with animal resistomes and microbiomes, as well as other researchers, including microbial ecologists and genomicists. I enjoyed reading it, and I believe the core technical analysis (on which all of the important biology in this type of manuscript rests) has been carefully designed and performed. Also, I have to congratulate the authors on the excellent and self-explanatory figures they designed, which unfortunately is not always the case. I have read the manuscript thoroughly and have only a few suggestions for the authors.

Main points:

It is unfortunate that there are no studies that the authors can use for relevant comparisons using poultry and pig samples from other countries besides Ghana and the 9 EU countries. It is certainly understandable that comparisons using metagenomic samples from different studies will introduce a significant bias rendering it impossible to draw any solid conclusions. However, maybe the authors could discuss a little more results from previous metagenomic studies in the resistome of poultry and pigs from other parts of the world, especially since Africa is considered a continent with the lowest antimicrobials usage in the world.

“We thank the reviewer for their general positive feedback. We agree with the reviewer that the comparison part of our study could be expanded a bit. We therefore have discussed our output in comparison not only to those 9 European countries, but also compared to the metagenomic study of all available shotgun metagenomic sequencing data from NCBI SRA by Lawther et al, 2022 (doi: 10.3389/fmicb.2022.897905) (Line 414). We would like to stress, as mentioned in the manuscript, the lack of data on such topic from Africa, therefore we are constrained by limiting the comparison and discussion to non-African regions.”

Throughout the figures and the manuscript text, there are interesting comparisons between the resistomes of the EU countries and the industrialized, rural, and urban resistomes of Ghana, but there is no info about the EU countries' resistomes. Were they collected from industrialized facilities or ecological farms? Considering the details about the Ghana sampling, the manuscript would benefit greatly if the authors provided some details about the origin of the EU samples. I understand these details may exist in the cited literature (and the effort website), but it would be nice if that information is also readily available in the manuscript. In short, this information should include the type of facilities (free range, industrialized etc), the type of feed (whenever possible), and the antibiotics usage (again, depending on the availability of information). If the metagenomic samples from the EU countries came from different types of facilities, maybe, it would make sense to separate and present these samples to industrialized/rural/urban, similarly to what was done for the Ghana samples.

“We thank the reviewer for their comment. We have now added more details about farm animals and their details in the EFFORT samples (Lines 146-147).”

Minor comments:

L79: do the authors have any details about the age of the animals?

“Farmers do not keep records of age and it was therefore not available. However, we ensured that all categories (piglets, weaners, growers, and adults) were sampled on each farm if available.”

L83: are any details available about the composition of the commercial feed and how it may differ from the European?

“Home-prepared feed for pigs, wheat bran (60%), soybean, maize chaff, and fish in proportions that are dependent on the availability of feed ingredients and the budget of the farmer.”

L89: do you know how frequently they were allowed to scavenge outside?

“For rural pigs, they scavenge till evening until the time when owners drive them back into their pens.”

L92: what does "self-medication" entail?

“When farmers treat the sick or give prophylactic medication to animals without recourse to trained veterinary personnel.”

L96: collected in sterile containers? falcon tubes?

“All samples were collected in a sterile fashion in sterile falcon tubes. The manuscript has been adjusted accordingly (Line 121).”

L102: please remove "apparently"; either they were purchased from the markets or not

“We thank the reviewer for commenting on this, as we realize this sentence could be misleading. The “apparently” in the sense was meant to refer to the health status of the chickens and not if they were purchased or not. We realize this sentence could be misleading and have rephrased it (Line 102).”

L107-109: sentence is a bit too colloquial, can you replace it with something like "Slaughtered birds arrive in central markets via intermediaries, who typically transport them from rural farms."

“We thank the reviewer for pointing this out. We have adjusted the sentence (Line 110-111).”

L115: please specify how long the interruptions were

“After collecting the first and second batches, the third batch was delayed for a week. The fourth batch was collected a month later; therefore, the number had to be increased to make up for time lost. Finally, the last batch was collected after seven weeks, with the number increased to complete the sampling. This has now been included in the manuscript.”

L119: this is the first mention, please specify which nine countries

“The manuscript has been adjusted accordingly (Line 142-143).”

L159: please specify which versions or when were the databases downloaded

“We thank the reviewer for this observation. The download date has been added to each database mention (Line 167).”

L207: I may have missed that, but how exactly did you estimate the number of genera per sample? Based on the 16S? Because, if yes, that is not ideal.... (I would suggest sth like MicrobeCensus for such estimations)

“We thank the reviewer for their comment. Yes, the number of unique genera per sample was estimated based on 16S. We realize that it is not ideal, and we have added the average number of genome equivalents per sample identified with MicrobeCensus to the manuscript (Line 218).”

L410-411: My point again, regarding more information about the diets (whenever possible)

“Unfortunately this is not available.”

Reviewer #2 (Comments for the Author):

In their manuscript "The resistome of rural and urban pigs and poultry in Ghana", the authors determine metagenomic data in livestock from Sub-Saharan Africa with a focus on antimicrobial resistance genes and bacterial composition. The authors identify differences between livestock microbiota in Ghana and Europe, which contributes to a better understanding of the global microbiome diversity and warrants further investigation to complete a microbiota catalogue encompassing livestock species and breeds, countries, and different production systems. The data are solid, well-presented and support the conclusions drawn by the authors. I have only a few comments to the manuscript.

Major points

Results, lines 269, 270: tetA(P) and tetB(P) are frequently associated with Clostridium and bla_{ACI} also with anaerobic bacteria. Can the authors comment on this, as Clostridia were identified as being among the more abundant genera in the pig samples (Table S4)?

“We thank the reviewer for their general positive feedback and their comment. As we already mention in the discussion the bacterial composition can have a major effect on the resistome. As correctly observed by the reviewer Clostridium was more abundant in pigs from Ghana compared to Europe. However, the higher abundance of tetA(P) and tetB(P) was when comparing pigs and poultry from Ghana and with a higher abundance in poultry. We do agree that further studies into co-abundance and gene synteny are needed, but would at this stage prefer not to become too speculative regarding two genes only.”

Discussion, Effects of host breed and diet on gut microbiome community and resistome: The bacterial composition and the resistomes differ strongly between pigs and poultry in Ghana and between the Ghanaian samples and the European samples. Differences in diet and in breeds present in the animal populations surely contribute the differences in microbiota and resistomes. Due to the means of poultry acquisition, there is no metadata as to antimicrobial usage and diet. This should be mentioned as a limitation to the study. Future studies should try to include such data.

“We thank the reviewer for their comment, and we agree such metadata should be included. We have tried to address this in the discussion.”

Discussion: Was or will a metabolic reconstruction of the metagenome be attempted. This is beyond the scope of this study, but it would be interesting to compare metabolic profiles of European and Ghanaian samples to identify differences and try to backtrace dietary influences.

“We thank the reviewer for this very valid comment. While metabolite predictions and functional analyses are extremely interesting and important for those data (metagenomics), we believe this is out of the scope of our current manuscript (as mentioned by the reviewer). However, we completely agree that it would be extremely interesting to compare how such changes between the locations and the animal populations might also translate into various functions (metabolomics) to show the adaptations to different area/locations. This is particularly interesting to investigate the functional differences between the bacterial taxa that are differently abundant between the populations like Enorma and

Subdoligranulum). However, this is outside the current manuscript focus and aim, but we do plan to carry out such analyses in combination with other datasets we currently have.”

Minor points

Title: Since the authors stress that the pig and poultry metagenomes differ strongly in bacterial composition and in antibiotic resistance genes from each other, they might want to change the title to "The resistomes of rural and urban pigs and poultry in Ghana".

Good point. has been done.”

Introduction, line 56: change spacity to sparsity or to scarcity.

“The manuscript has been adjusted.”

M&M, Pig samples, lines 84-87: change Netting to netting, Veterinary to veterinary and medications to medications.

“The manuscript has been adjusted.”

M&M, Poultry samples: were the chickens from rural and urban sources also layers or broilers? Is there any information on the chicken breed or breeds that were slaughtered?

“All the rural and urban chickens were local (Ghanaian) breeds. The industrial chickens were foreign breeds, though the exact breed is not known.”

M&M, Antimicrobial resistance gene quantification, line 154: which scheme was used to classify the resistance genes?

“Classification of resistance genes was made from the official annotation given by the ResFinder database documentation (https://bitbucket.org/genomicepidemiology/resfinder_db/src/master/). We have added this information to the manuscript (Line 161-162).”

Results, Summary of the data, line 203: the standard deviation of the sample is quite high, although there is only one sample each with a very low and one with a very high sample count. It might be easier to evaluate the homogeneity of the sample if a size distribution plot of the fragment counts were given in the supplemental material.

“We thank the reviewer for this suggestion. A size distribution plot of the fragments counts has been added to the manuscript (Figure S2).”

Results, line 267: change resistant to resistance.

“The manuscript has been adjusted (line 277).”

Results, line 292 replace cfc(C) with cfr(C).

“The manuscript has been adjusted (Line 281).”

August 2, 2023

Prof. Frank M. Aarestrup
Danmarks Tekniske Universitet
National Food Institute
Kemitorvet
Lyngby 2800
Denmark

Re: mSystems00629-23R1 (The resistomes of rural and urban pigs and poultry in Ghana)

Dear Prof. Frank M. Aarestrup:

Your manuscript has been accepted, and I am forwarding it to the ASM Journals Department for publication. For your reference, ASM Journals' address is given below. Before it can be scheduled for publication, your manuscript will be checked by the mSystems production staff to make sure that all elements meet the technical requirements for publication. They will contact you if anything needs to be revised before copyediting and production can begin. Otherwise, you will be notified when your proofs are ready to be viewed.

If you would like to submit a potential Featured Image, please email a file and a short legend to msystems@asmusa.org. Please note that we can only consider images that (i) the authors created or own and (ii) have not been previously published. By submitting, you agree that the image can be used under the same terms as the published article. File requirements: square dimensions (4" x 4"), 300 dpi resolution, RGB colorspace, TIF file format.

We recognize that the video files can become quite large, and so to avoid quality loss ASM suggests sending the video file via <https://www.wetransfer.com/>. When you have a final version of the video and the still ready to share, please send it to mSystems staff at msystems@asmusa.org.

Sincerely,

Rachel Poretsky
Editor, mSystems

Journals Department
E-mail: mSystems@asmusa.org